# Effectiveness of COVID-19 Vaccines in People with Severe Mental Illness: A Systematic Review and Meta-Analysis

**DOI:** 10.3390/vaccines12091064

**Published:** 2024-09-18

**Authors:** Wen Dang, Iman Long, Yiwei Zhao, Yu-Tao Xiang, Robert David Smith

**Affiliations:** 1Unit of Psychiatry, Department of Public Health and Medicinal Administration, Institute of Translational Medicine, Faculty of Health Sciences, University of Macau, Macao SAR, China; yc27639@umac.mo (W.D.); yc27652@umac.mo (I.L.); mc24741@connect.um.edu.mo (Y.Z.); ytxiang@um.edu.mo (Y.-T.X.); 2Centre for Cognitive and Brain Sciences, University of Macau, Macao SAR, China

**Keywords:** severe mental illness, schizophrenia, bipolar disorder, major depressive disorder, COVID-19 vaccines

## Abstract

Prior to the introduction of COVID-19 vaccines, patients with severe mental illness (SMI) were at greater risk of COVID-19-related outcomes than the general population. It is not yet clear whether COVID-19 vaccines have reduced the risk gap. We systematically searched nine international databases and three Chinese databases to identify relevant studies from December 2020 to December 2023 to compare the risk of COVID-19-related outcomes for SMI patients to those without SMI after vaccination. Random effects meta-analysis was performed to estimate the pooled odds ratio (OR) with 95% confidence intervals (CI). Subgroup analysis, sensitivity analysis, and publication bias analysis were conducted with R software 4.3.0. A total of 11 observational studies were included. Compared with controls, SMI patients were associated with a slightly increased risk of infection (pooled OR = 1.10, 95% CI, 1.03–1.17, *I*^2^ = 43.4%), while showing a 2-fold higher risk of hospitalization (pooled OR = 2.66, 95% CI, 1.13–6.22, *I*^2^ = 99.6%), even after both groups have received COVID-19 vaccines. Limited evidence suggests a higher mortality risk among SMI patients compared to controls post vaccination, but the findings did not reach statistical significance. SMI patients remain at increased risk compared to their peers in COVID-19-related outcomes even after vaccination. Vaccination appears an effective approach to prevent severe COVID-19 illness in SMI patients, and actions should be taken by healthcare providers to improve vaccination coverage in these vulnerable groups.

## 1. Introduction

Severe mental illness (SMI) refers to individuals with psychological problems that are so debilitating they affect their ability to engage, function within society, and severely impede their ability to take up occupational activities [1]. Although the definition of SMI varies across different studies, the most common conditions of SMI include schizophrenia (SZ) and related psychotic disorders, bipolar disorder (BD), and major depressive disorder (MDD) [2,3,4,5,6]. The prevalence of SMI varies in countries due to heterogeneity in methodological approaches, accounting for 5.5% in the United States (US) [7], 5% in Australia [8], and 1% in the United Kingdom (UK) [9]. Estimations on the global prevalence of SMI diseases show an increasing burden over time [10].

Physical health inequalities exist in the patients with SMI compared with the general population. Evidence has consistently demonstrated approximately 2–3 times the mortality risks in patients with SMI compared with those without SMI [11], with 10–25 years of potential life lost [12,13]. The COVID-19 pandemic has further exacerbated existing health disparities and placed a significant burden on this vulnerable population. Compared to the general population, SMI patients tend to suffer poorer physical conditions, shorter life expectancy, and excessive mortality rates [14,15]. Different studies across different countries have identified a higher risk of more severe COVID-19 infection, hospitalization, and mortality in people with SMI compared to the general population [2,16,17,18,19,20,21]. 

Once widely available, vaccination was recommended as the most effective and cost-effective strategy for reducing COVID-19 infections, severity, and mortality [22,23,24,25,26]. With the advocacy of prioritizing vaccination for vulnerable groups, Denmark, Germany, Netherlands, and the UK implemented policies for SMI patients to be prioritized for vaccination [27,28,29,30,31]. 

Concerns have arisen regarding the effectiveness and duration of protection provided by first-generation COVID-19 vaccines, due to the emergence of new SARS-CoV-2 variants and a decreasing trend in antibody titers in vaccinated individuals over time [32]. In addition to the increased risk in SMI patients of COVID-19 infection, hospitalization, and mortality, SMI patients are also at lower levels of protection against newer virus variants compared to the general population [33]. Given the increased risk of poor COVID-19 outcomes and possible reduction in vaccine effectiveness for SMI patients, it is important for health providers to have a comprehensive understanding of the effectiveness of COVID-19 vaccines in this group. 

It remains unclear if COVID-19 vaccines have contributed to reducing physical health inequalities gap for patients with SMI, with different evidence appearing among studies that are heterogeneous in their methodological approaches. To address this research gap, we conducted a systematic review and meta-analysis to evaluate the evidence related to the effectiveness of the COVID-19 vaccine in SMI patients compared to non-SMI controls. 

## 2. Methods

This systematic review is reported following the Meta-analysis of Observational Studies in Epidemiology (MOOSE) checklist [34] and the Preferred Reporting Items for Systematic Reviews and Meta-Analyses (PRISMA) statement [35]. The study protocol was registered in the PROSPERO (Registration number: CRD42022360844).

### 2.1. Search Strategy

To ensure that no relevant studies were missed, our search strategy not only included peer-reviewed published articles, but also actively tracked preprint databases. This allowed us to capture and incorporate any potential studies that might be published in the future. Systematically searches were executed in nine international databases (PubMed, EMBASE, CINAHL Ultimate, PsycINFO, Web of Science, the World Health Organization COVID-19 Database, Cochrane Database of Systematic Reviews, MedRxiv, BioRxiv) and three Chinese databases (CNKI, WanFang, and SinoMed). We restricted articles to those published after 1 December 2020, as December 2020 was the earliest time at which vaccination programs were initiated at the population-level [31,36]. The final search date was on 14 December 2023. The search terms were developed to be highly sensitive to identify all studies relevant to the key concepts of COVID-19 vaccination and SMI (Appendix A). Hand-searching was conducted in Google Scholar and in the reference lists of included articles to ensure no omissions.

### 2.2. Inclusion and Exclusion Criteria

All retrieved articles were screened using the web-based software Rayyan [37]. Using Rayyan facilitated the screening and selection of relevant studies, streamlining the process and enhancing efficiency. Each retrieved study was independently screened by at least two reviewers, first screening titles and abstracts and successively full texts. Study inclusion criteria was based on the PICOS strategy as follows: Participants (P): patients with a broad definition of severe mental illness, including a diagnosis of schizophrenia, schizoaffective disorder, bipolar disorder, major depressive disorder, or other psychotic disorders; patients aged 18 or above; Intervention (I): COVID-19 vaccines approved by the WHO [38], including inactivated vaccine, protein-based vaccines, viral vector vaccines, RNA or DNA vaccines; Controls (C): individuals without a diagnosis of SMI. Outcome (O): the cases of COVID-19 infections, COVID-19-related hospitalization, and mortality after vaccination; Study design (S): all observational studies and randomized controlled trials were eligible. Exclusion criteria included cross-sectional design studies, reviews, comments, editorials, conference papers, case-report papers, or animal studies. Publications not in English or Chinese were excluded. Researchers were blinded to each other’s decisions. Any disagreement was discussed within the whole review team after unblinding.

### 2.3. Data Extraction and Quality Assessment

Data extraction was conducted by two reviewers independently, using a standardized form created for the purposes of this review; the information extracted included the first author, publish year, study design, data resources, sample characteristics in the SMI group and non-SMI group, type of the COVID-19 vaccine, outcome of interest, and follow-up period. The missing data or unreported data was filled in by contacting study authors to provide additional details. The Newcastle–Ottawa scale (NOS) was used to assess the methodological quality of the observational study from three perspectives, including selection (4 stars), comparability (2 stars), and exposure/outcome (3 stars), with the highest quality studies achieving 9 stars [39,40]. Any discrepancies between reviewers were solved by consulting with a senior researcher with extensive experience in conducting systematic reviews and meta-analyses in our research team.

### 2.4. Statistical Analysis

Odds ratio (OR) with 95% confidence intervals (CIs) were calculated as the effect size for the meta-analysis of outcomes. Studies reporting multiple SMI subgroups compared with the same control group were included in the meta-analysis by combining the subgroups to create a single SMI group. The potential sources of heterogeneity between studies were considered prior to analysis as not being limited to sampling; a random effects model based on an inverse–variance approach was used. An *I*^2^ of at least 50% was considered to represent substantial heterogeneity. Subgroup analyses were performed by different variables when there was an adequate number of relevant studies. Leave-one-out analysis was conducted to test the robustness of the result of the meta-analysis. The risk of publication bias was estimated by funnel plot and Egger’s test to estimate the likelihood of funnel plot asymmetry. The statistical analysis was conducted by R version 4.3.0 [41] with the *meta, metafor* package. Two sides of *p* values less than 0.05 determined significance.

## 3. Results

### 3.1. Search Results and Recruited Studies

Figure 1 shows the flow chart of the study selection. A total of 4009 articles were retrieved, with 3869 articles from international databases and 140 articles from Chinese-based databases. A total of 639 duplicate records were removed before the screening. Of 3370 abstracts and titles screened, 179 (4.5%) were relevant for full-text review. Four articles’ authors responded to provide detailed data that could be used in this meta-analysis [42,43,44,45]. Finally, 11 articles (0.61%) reporting on 1,592,778 SMI patients and 27,962,772 vaccinated controls were included for meta-analysis [42,43,44,45,46,47,48,49,50,51,52].

### 3.2. Characteristics of the Included Studies

All the detailed information for the included studies is listed in Appendix A. Five studies were from European countries, two from North American countries, and four from Asian countries. Nine out of eleven studies collected data from large electronic medical records, and two studies in China (mainland and Taiwan) collected data from multiple hospitals and a single medical center. Cohort study designs were conducted by eight studies, and three studies used a case-control study design. The definition and subtype of SMI populations varied among the included studies, which could be divided into three main categories, including SZ group (patients with schizophrenia, psychosis, and other psychotic disorders), BD group (including patients with bipolar disorders) and MDD group (including patients with major depressive disorders). Across all studies, nine studies reported different types of COVID-19 vaccines including mRNA, vector viral, inactivated and protein subunit vaccines, with two studies not reporting detailed information on vaccine type. Three out of eleven studies evaluated the impact of the booster dose of vaccines on severe COVID-19 illness or death, while most did not report outcomes according to the types and dosages of vaccines used.

### 3.3. Quality Assessment of Included Studies

The total NOS scores for 11 studies ranged from 7 to 9, averaging 8.36, indicating moderate to high methodological quality (Appendix A). A total of 8 out of the 11 studies rated high on the domain of selection, as these studies derived from the electronic medical record based on national-level data, suggesting the representativeness of the population. In the domain of comparability, all studies received at least a score of 1, indicating the adequacy of controlling for potential confounders. Additionally, nine studies were evaluated with the highest score in reporting outcomes, while the Tzur Bitan et al. [46] and Zhu et al. [50] studies were assessed with lower scores due to the insufficient length of follow-up for outcomes to occur.

### 3.4. Outcomes

The results of the primary analysis for each COVID-19 outcome are listed in Table 1.

The pooled absolute rate of COVID-19 infection, hospitalization, and mortality post vaccination in the SMI group was 2.60% (95% CI: 0.22–24.05), 2.60% (95% CI: 0.24–22.93) and 0.05% (95% CI: 0.01–0.37), respectively (Table 1). SMI patients were at a slightly increased risk of COVID-19 infection (OR = 1.10, 95% CI, 1.03–1.17, *p* < 0.01, *I*^2^ = 43.4%) and hospitalization (OR = 2.66, 95% CI 1.13–6.22, *p* = 0.02, *I*^2^ = 99.6%) compared to the control group. However, the pooled estimates showed an increased but not significant risk of mortality among vaccinated SMI patients than non-SMI control (OR = 2.67, 95% CI 0.27–26.45, *p* = 0.40, *I*^2^ = 98.3%). The *I*^2^ statistic and forest plot indicated substantial and high heterogeneity across studies (Figure 2).

For the outcome of COVID-19 infection, the subgroup analysis by different SMI subtypes showed that a slightly increased risk was observed in the SZ group (OR = 1.06, 95% CI, 1.01–1.12, *p* = 0.03, *I*^2^ = 0.0%) and MDD group (OR = 1.10, 95% CI, 1.02–1.18, *p* < 0.05, *I*^2^ = 0.0%) when compared with control. SMI patients showed a reduced risk of infection in two studies from Asia (OR = 0.29, 95% CI, 0.10–0.84, *p* = 0.02, *I*^2^ = 0.0%). In contrast, two studies from North America reported an increased risk of infection in SMI patients compared to those without SMI (OR = 1.13, 95% CI, 1.10–1.15, *p* < 0.001, *I*^2^ = 0.0%). The subgroup analysis by study design was only conducted in five cohort studies; the consistent results added confidence to our main findings (OR = 1.11, 95% CI 1.09–1.13, *p* < 0.001, *I*^2^ = 0.0%).

For the outcome of COVID-19-related hospitalization, the subgroup analysis by subtype showed patients with SZ (OR = 3.27, 95% CI 1.21–8.84, *p* = 0.02, *I*^2^ = 99.3%) and BD (OR = 1.97, 95% CI 1.53–3.89, *p* < 0.001, *I*^2^ = 0.0%) had significantly increased risks of hospitalization than controls. In subgroup analysis by location, three studies from Europe showed an increased pooled effect size of hospitalization in SMI patients compared to those without SMI (OR = 4.04, 95% CI 1.01–16.21, *p* = 0.05, *I*^2^ = 99.9%). In subgroup analysis by study design, robust results were identified in two different designs.

For the outcome of COVID-19-related hospitalization, the result of the meta-analysis did not show a statistically significant difference between groups (Table 2).

### 3.5. Sensitivity Analysis and Publication Bias

The leave-one-out analysis showed the lowest *I*^2^ was achieved when excluding “Tzur Bitan et al., 2021 [46]” and “Semenzato et al., 2022 [47]” in the meta-analysis for the result of infection and mortality, respectively, but without changing the direction of results. The analysis showed no significant change in the results of hospitalization. The forest plot (Appendix A) showed the recalculated pooled effects after each study was omitted. Although there was an observed asymmetry from the funnel plot for each outcome (Appendix A), no significant publication bias was detected based on the result of Egger’s test (Appendix A). However, the interpretation of the funnel plots should be approached cautiously, as limited articles were included.

## 4. Discussion

To the best of our knowledge, this is the first systematic review and meta-analysis comparing the effectiveness of COVID-19 vaccinations between SMI groups and non-SMI groups. The results of the meta-analysis indicated that SMI patients remain at higher risk of COVID-19-related outcomes in comparison to individuals without SMI, but the pooled absolute risk for infection or infection-related hospitalization after vaccination was rare for people with SMI post vaccination. No significant increased risk of mortality was identified among vaccinated SMI patients than non-SMI controls. Although there was a low absolute rate of mortality in SMI patients post vaccination, the findings might be attributed to potential bias, which should be illustrated with caution.

The rates of infection and hospitalization after vaccination appear to be lower compared to other studies that estimated risks in this population before vaccination. The absolute rates of COVID-19-related infection and hospitalization in SMI patients post vaccination across all studies were estimated at 2.6%, which was lower than the reported percentage of patients with SMI without vaccination who tested positive for SARS-CoV-2 antibodies in the cross-sectional study conducted in Denmark [53]. Patients with schizophrenia and BD had over a three-fold increase in the odds of hospitalization pre vaccination compared to those without SMI [2], which seemed higher than the risk of hospitalization from our findings among these patients post vaccination. Additionally, the low absolute mortality rate in the SMI group might indicate the effectiveness of vaccination in minimizing the disparities between SMI patients and the general population. However, the potential selection bias may underestimate the risk of death post vaccination, as SMI patients who died before vaccination could not be taken into consideration.

Despite the protection of vaccines, individuals with SMI still have an increased residual risk to COVID-19 compared with the general population. Poor physical health and behavioral risk factors are potential contributing factors that may explain the higher risk of COVID-19-related outcomes after vaccination in SMI patients [54]. For patients with SMI, the use of antipsychotic medication, higher rates of poor dietary intake, and a high prevalence of smoking may contribute to lower overall physical health conditions. A higher prevalence of co-morbid physical disease compared with the general population may also contribute to the increased risk [15]. The possible pathophysiological mechanism of how physical diseases influence COVID-19 outcomes is the hyperactivation of the immune system. Having SMI with comorbidities is frequently associated with a pro-inflammatory state. When an individual with these conditions contracts SARS-CoV-2, the inflammatory state may exacerbate tissue damage, lead to multi-organ failure, and potentially fatal outcomes [55]. Additionally, a high prevalence of sleep difficulties reported in SMI patients [56], could be another potential mechanism, as poor sleep quality substantially impairs immune function [57]. Combinations of these contributing factors may lead to immune dysregulation, consequently influencing the immune response to vaccines, and SMI patients’ overall immune ability to act against a COVID-19 infection.

Although these underlying health conditions of SMI explain the increased risk of more severe courses of COVID-19 and reduce the effectiveness of vaccines received, the continuous evolution of the COVID-19 virus also limited the effectiveness of vaccination. With the continued risk of COVID-19 infections, at the time of conducting this review, five identified variants which were classified as variants of concern, labeled as Alpha, Beta, Gamma, Delta, as well as Omicron, respectively [58]. The consistent results suggested that the virus variants are associated with the effectiveness of vaccines [23,58]. A full vaccination of COVID-19 vaccines showed high effectiveness against the Alpha variant, and moderate effectiveness against the Beta, Gamma, and Delta variants [23,58], while booster vaccination was found more effective against the Delta and Omicron variants [58]. However, lower vaccination uptake rates of initial and booster vaccination were identified in SMI patients in Israel [59] and France [60]. The WHO statement on ending COVID-19 as a public health emergency does not mean this disease is not a global threat [61]. Increasing the promotion of full vaccination regimens, including booster doses, remains a public health priority for people with SMI. In addition, the effectiveness of different types of COVID-19 vaccines varies in preventing COVID-19 infection. Although we collected data from included studies on vaccinated individuals with SMI, the outcomes related to the specific type of vaccination received were not reported sufficiently, which hindered our ability to conduct a subgroup analysis. In a summary of data from the real world, mRNA vaccines displayed higher effectiveness against SARS-CoV-2 infection and variants [58,62].

Notably, we observed that SMI patients had a significantly lower risk of COVID-19 infection compared to non-SMI controls in the Asian population, which was consistent with the findings from the study conducted in South Korea [63]. This is possibly attributed to the included studies with short time follow-ups and limited sample sizes of the Asian population. Additionally, a wider confidence interval of the results in Asian countries was observed than the study results in Western countries, indicating the need for further improvement in study quality. Furthermore, the specific biological mechanisms underlying the ethnically specific effect of COVID-19 vaccination on COVID-19-related outcomes for individuals with SMI require further investigation.

There were several limitations in our review. First, despite the majority of studies being based on national electronic health records [42,43,46,47,48,49,51] providing a typically large sample size with real-world clinical data, these records might cause bias in estimating the outcomes, for example, the potential of underestimation of the population with COVID-19 infection, as these infections were limited to those recorded by healthcare. Other issues for these data could also include the incomplete and inadequate capturing of secondary care information and missing data. In contrast, primary data collection may provide more detailed information but with a limited achievable sample size [44,50]. Additionally, the target sample in one study was focused on veterans with an unbalanced distribution of males and females, which may account for the lack of generalizability [42]. Second, some of the planned subgroup analyses were not conducted due to a lack of sufficient reported data for analysis; an example of this was the underreporting of vaccine types. Future studies may focus on the evaluation of the effectiveness of different vaccine types and dosages among SMI patients. Third, we only evaluated the physical COVID-19-related outcomes in SMI patients and did not take into consideration if there were any psychiatric adverse events following COVID-19 vaccination. Recent evidence from a large population-based cohort study in South Korea has shown that COVID-19 vaccination may increase the risks of certain psychiatric disorders, including depression, anxiety, and sleep disorders, while potentially reducing the risk of conditions like schizophrenia and bipolar disorder [64]. Evaluating the impact of COVID-19 vaccination on psychiatric outcomes in SMI patients can help to provide a comprehensive perspective on the effectiveness of COVID-19 vaccines. Fourth, due to the relatively short observation periods of the included studies, very few were able to report on the outcome of COVID-19-related mortality. In the future, prolonging the observation period will help capture the occurrence of mortality events. Lastly, eight studies were conducted in high-income countries, which may prevent the result from being generalized to lower-income countries.

Medical specialists, especially psychiatrists, play a positive role in addressing vaccine hesitancy or refusal from mental illness patients and ensuring equitable access to the COVID-19 vaccine [65]. They can provide SMI patients with adequate information, respectfully address negative attitudes, and discuss the advantages and possible risks of vaccination [29]. Our findings can give these healthcare providers confidence in improving vaccination coverage within this vulnerable group. Although vaccination has substantially reduced the risk of COVID-19-related outcomes, these risks persist in patients with SMI. Proactively communicating with SMI patients and their caregivers about the increased risk for those patients can encourage further preventive steps for infection or monitoring for severe illness if infected.

## 5. Conclusions

This systematic review and meta-analysis of 11 observational studies involving 1,592,778 SMI patients and 27,962,772 controls across six countries showed the COVID-19 vaccines are effective against severe illness for patients with SMI. Despite the protection of vaccines, individuals with SMI remain at a higher risk of contracting the infection and developing severe illness post vaccination compared with controls. Health professionals should be aware of the risk disparity between SMI patients and non-SMI individuals and take proactive steps to mitigate health inequality among SMI patients.

## Figures and Tables

**Figure 1 vaccines-12-01064-f001:**
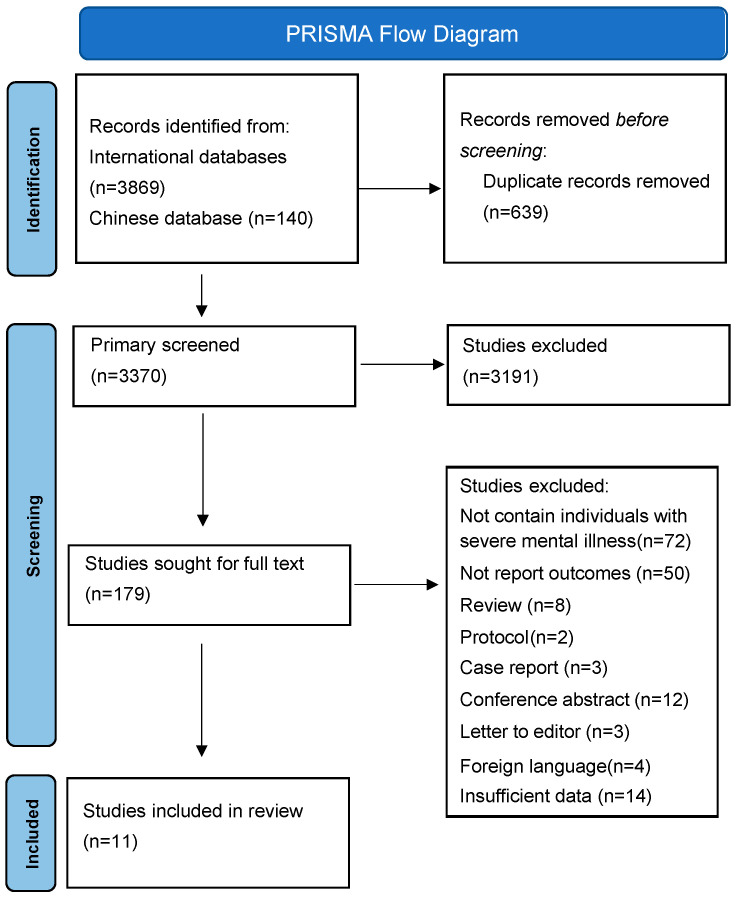
PRISMA flow diagram.

**Figure 2 vaccines-12-01064-f002:**
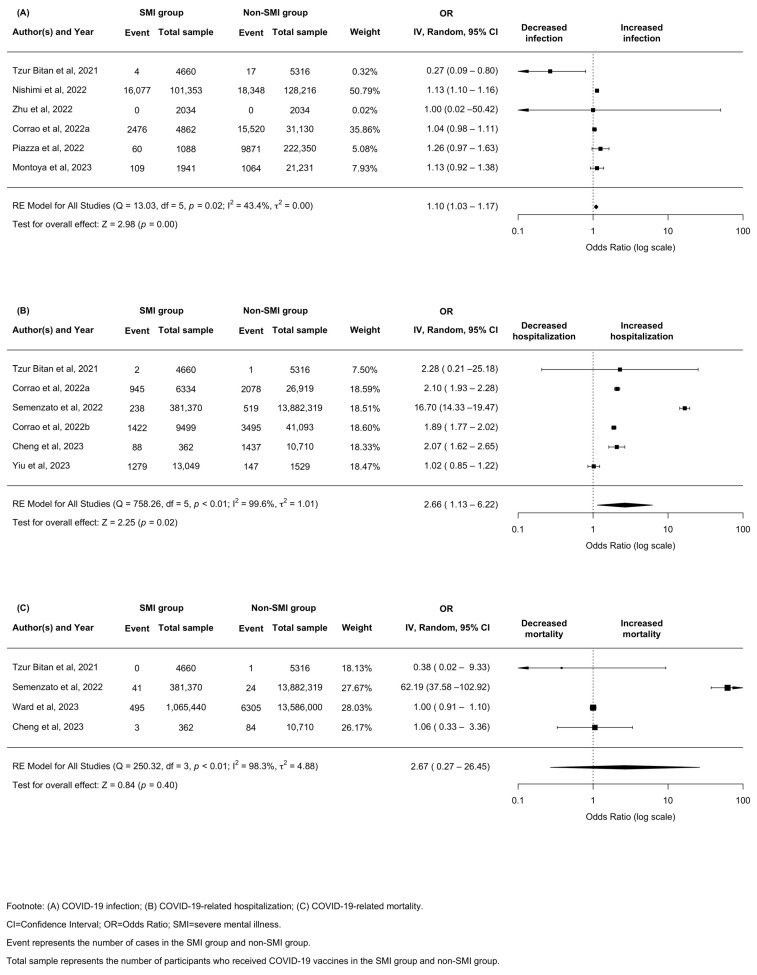
Forest plot illustrating odds ratio of COVID-19-related outcomes post vaccination in patients with severe mental illness [42,43,44,45,46,47,48,49,50,51,52]. Footnote: (**A**) COVID-19 infection; (**B**) COVID-19-related hospitalization;(**C**) COVID-19-related mortality. SMI = severe mental illness; CI = confidence interval; OR =odds ratio. Event represents the number of cases in the SMI group and non-SMI group. Total sample represents the number of participants who received COVID-19 vaccines in the SMI group and non-SMI group.

**Table 1 vaccines-12-01064-t001:** Result of the meta-analysis for COVID-19 related outcomes.

Outcome	Studies	SMI Group			Non-SMI Group			Estimates	
		Event	Total Sample Size	Proportion (%)	Event	Total Sample Size	Proportion (%)	Pooled OR	*I*^2^ (%)
Infection	6	18,726	115,938	2.60 (0.22–24.05) **	44,820	410,277	3.03 (0.35–21.87) **	1.10 (1.03–1.17) **	43.4
Hospitalization	6	3974	415,274	2.60 (0.24–22.93) **	7677	13,967,886	1.02 (0.05–16.76) **	2.66 (1.13–6.22) *	99.6
Mortality	4	539	1,451,832	0.05 (0.01–0.37) ***	6414	27,484,345	0.02 (0.00–0.61) ***	2.67 (0.27–26.45)	98.3

Footnotes: * means *p* < 0.05; ** means *p* < 0.01; *** means *p* < 0.001. SMI = severe mental illness. OR = odds ratio.

**Table 2 vaccines-12-01064-t002:** Meta-analysis of COVID-19-related outcomes comparing SMI patients and non-SMI patients in subgroups.

Subgroup	Studies (*n*)	SMI Group (*n*)	Non-SMI Group (*n*)	Pooled OR	*I*^2^ (%)
COVID-19 infection					
SMI subtype					
SZ group	6	16,811	410,277	1.06 (1.01–1.12) *	0.0
MDD group	3	89,037	161,380	1.10 (1.02–1.18) *	62.8
BD group	3	10,090	161,380	0.99 (0.63–1.57)	45.8
Study location					
Asia	2	6694	7350	0.29 (0.10–0.84) *	0.0
Europe	2	5950	253,480	1.09 (0.93–1.28)	45.6
North America	2	103,294	149,447	1.13 (1.10–1.15) ***	0.0
Study design					
Cohort	5	111,076	379,147	1.11 (1.09–1.13) ***	0.0
COVID-19-related hospitalization					
SMI subtype					
SZ group	5	394,155	139,577,176	3.27 (1.21–8.84) *	99.3
MDD group	3	20,201	69,541	1.46 (0.96–2.23)	97.9
BD group	2	556	42,622	1.97 (1.53–3.89) ***	0.0
Study location					
Asia	3	18,071	17,555	1.49 (0.80–2.78)	89.5
Europe	3	397,203	13,950,331	4.04 (1.01–16.21) *	99.9
Study design					
Cohort	3	386,392	13,898,345	4.88 (1.07–22.23) *	98.6
Case-control	3	28,882	69,541	1.61 (1.04–2.48) *	98.4
COVID-19-reltaed mortality					
SMI subtype					
SZ group	2	386,030	13,887,635	6.28 (0.04–904.06)	89.5
Study location					
Asia	2	5022	16,026	0.94 (0.32–2.79)	0.0
Europe	2	1,446,810	27,468,319	7.83 (0.14–447.74)	99.6
Study design					
Cohort	4	1,451,832	27,484,345	2.67 (0.27–26.45)	98.3

Footnotes: * means *p* < 0.05; *** means *p* < 0.001. OR = odds ratio. SMI = severe mental illness; SZ group means patients with schizophrenia, psychosis, and psychotic disorders; BD group means patients with bipolar disorder; MDD group means patients with major depressive disorder.

## Data Availability

The authors confirm that the data supporting the findings of this study are available within the article and the Appendix A. Raw data and code that support the findings of this study are available from the corresponding author, upon reasonable request.

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
