# Peer review of "Effectiveness of COVID-19 Vaccines in People with Severe Mental Illness: A Systematic Review and Meta-Analysis"

_vaccines, 2024, doi:10.3390/vaccines12091064_

Round 1
Reviewer 1 Report
Comments and Suggestions for Authors
Dear authors,
Your manuscript “Effectiveness of COVID-19 Vaccines in People with Severe Mental Illness: A Systematic Review and Meta-analysis” describes a systematic review and meta-analysis performed in the study to evaluate the evidence related to the effectiveness of the COVID-19 vaccine in severe mental illness patients. In general, the manuscript is well written, and gives a lot of new and important information in a field of COVID-19 protection issues.
However, the manuscript revision is required.
First, English revision is required.
Second, I have the following remarks:
Throughout the text - “et al” – correct, please, to “et al.”;
L18-19 – “higher risk of hospitalization (pooled OR= 2.66, 95% CI, 1.13-6.22, I2=99.6%) postvaccination.” – what do you mean under “risk of hospitalization postvaccination”? It seems that it needs to be revised for clear understanding;
L73 – “MedRxiv, BioRxiv)” - I’m not sure that including the preprint databases (without peer review process) is a good idea for meta-analysis purpose; however you are free to explain it more clearly;
L194 – “did not yield significant results” - It is not clear what result should be considered significant? I recommend rephrase;
Table 2 – “COVID-19-reltaed” – correct, please.
Comments on the Quality of English LanguageThere are some errors in both grammar and punctuation.
Author Response
1. Summary
Thank you very much for reviewing this manuscript. We appreciate all your kind suggestions and thoughtful feedback, which will help us improve the quality of our work.
2. Point-by-point response to Comments and Suggestions for Authors
Comments 1: English revision is required.
Response 1: Thanks for your recommendation. We have carefully reviewed the manuscript for grammatical and punctuation errors and made the necessary corrections. This manuscript has been language revised by native speakers.
Comments 2: Throughout the text - “et al” – correct, please, to “et al.”
Response 2: Thank you for pointing this out. We have modified the formatting of "et al" throughout the text to "et al."
On page 5, lines 159-161, the sentence “Additionally, 9 studies were evaluated with the highest score in reporting outcomes while Tzur Bitan et al [46] and Zhu et al [50] were assessed with lower scores for the reason of the insufficient length of follow-up for outcomes to occur” is revised to “Additionally, 9 studies were evaluated with the highest score in reporting outcomes while Tzur Bitan et al. [46] and Zhu et al. [50] were assessed with lower scores for the reason of the insufficient length of follow-up for outcomes to occur”.
Comments 3: L18-19 – “higher risk of hospitalization (pooled OR= 2.66, 95% CI, 1.13-6.22, I2=99.6%) postvaccination.” – what do you mean under “risk of hospitalization postvaccination”? It seems that it needs to be revised for clear understanding;
Response 3: Thank you for the suggestion. We clarify that “risk of hospitalization postvaccination” refers to the increased risk of admission to hospital in individuals with SMI compared to the general population, after both groups have received COVID-19 vaccines. This finding highlights the persistent health disparities experienced by individuals with SMI, emphasizing the significance of the result and calling the need for targeted interventions to address their unique vulnerabilities.
On page 1, lines 17-20: To make this sentence more readable and clearer, we have changed this sentence into “Compared with controls, SMI patients were associated with a slightly increased risk of infection (pooled OR=1.10, 95%CI, 1.03-1.17, I2=43.4%), while showing a 2-fold higher risk of hospital admission (pooled OR= 2.66, 95% CI, 1.13-6.22, I2=99.6%), after both groups have received COVID-19 vaccines”.
Comments 4: “MedRxiv, BioRxiv)” - I’m not sure that including the preprint databases (without peer review process) is a good idea for meta-analysis purpose; however, you are free to explain it more clearly.
Response 4: We understand the reviewer's concern regarding the inclusion of preprint databases in our meta-analysis. To ensure a comprehensive search and minimize missing relevant studies, we included preprint databases in our search strategy, recognizing that some preprints may eventually be published in peer-reviewed journals. All included articles in our meta-analysis were peer-reviewed, to address this we have added the following.
On page 2, lines 73-76, we have added a paragraph to the Methods section to clarify our rationale for including these databases. The sentence "To ensure that no relevant studies were missed, our search strategy not only included peer-reviewed published articles, but also actively tracked preprint databases. This allowed us to identify any potential studies that might be published in the future." is added.
Comments 5: L194 – “did not yield significant results” - It is not clear what result should be considered significant? I recommend rephrase;
Response 5: Thank you for your suggestion. The original sentence was rephrased.
On page 7, lines 200-201, the revised sentence reads: " For the outcome of COVID-19-related admissions to hospital, the meta-analysis did not show a statistically significant difference between groups.
Comments 6: Table 2 – “COVID-19-reltaed” – correct, please.
Response 6: Thank you for pointing this out.
We have corrected the spelling error in Table 2, changing "COVID-19-reltaed" to "COVID-19-related."
3. Response to Comments on the Quality of English Language
Point 1: There are some errors in both grammar and punctuation.
Response 1:
Thank you for pointing this out. We have carefully reviewed the manuscript for grammatical and punctuation errors and made the necessary corrections.
On page 1, lines 30-32, the first sentence in the Introduction section has a grammatical error. The error lies in the pronoun "it". It refers to "psychological problems" which are plural. The correct pronoun to use is "they". Thus, the sentence “Severe mental illness (SMI) refers to individuals with psychological problems that are so debilitating it affects their ability to engage, function within society and severely impedes their ability to take up occupational activities” is revised to “Severe mental illness (SMI) refers to individuals with psychological problems that are so debilitating they affect their ability to engage, function within society and severely impede their ability to take up occupational activities”.
On page 2, lines 64-66, there is a grammar error in the sentence. We have corrected this sentence “To address this research gap, we conducted a systematic review and meta-analysis to evaluate the evidence related to the effectiveness of the COVID-19 vaccine in SMI patients compared to non-SMI controls.”
On page 3, line 126, we have revised the spelling error in the sentence “The statistical analysis as conducted by R version 4.3.0”, changing “as” to “was”.
On page 9, line 271, we have added a “the” before “promotion”.
Reviewer 2 Report
Comments and Suggestions for Authors
This study by Dang and colleagues used meta-analysis to evaluate numerous studies on patients with SMI and the effectiveness of the SARS-CoV-2 vaccine in patients with severe mental illness (SMI). SMI was defined as having either schizophrenia or other psychotic disorders (SZ), bipolar disorder (BD), or major depressive disorder (MDD). Patients were studied from December 2020 to December 2023. Three subgroups (SZ, BD, MDD) were also evaluated within these two groups. Overall, this study is well-written, the experimental design appropriate and showed that while SMI patients were associated with a slightly higher risk of infection, the data was not considered statistically significant. While the results are not particularly surprising since these patients are likely not immunocompromised, the study summarizes the numerous studies done to date and may be of to the field.
Comments on the Quality of English Language
Author Response
We appreciate the reviewer's positive assessment of our study, including the clarity of our writing and the appropriateness of our experimental design. We are grateful for their recognition of the study's contribution to the field. We believe that our study provides valuable insights into the nuances of vaccine effectiveness in SMI patients, particularly when considering the subgroup analyses. The findings might be of interest to researchers and clinicians working with this vulnerable population.Reviewer 3 Report
Comments and Suggestions for Authors
The topic of paper is of great interest. The paper is well structured and facilitates the reading and the follow-up of the study. However, some minimal clarifications must be included to accept their publication.
Methodologically the study is well supported and explained. The section of Materials and methods is very well explained. However, it would be interesting to include the keywords used in the search criteria to facilitate the reproducibility of the study.
In systematic reviews, data extraction is essential, as the accuracy of the data and its synthesis provide the basis for drawing appropriate conclusions. This is a slow and careful task that can later be converted and standardized so that no potentially relevant information is lost. In line 106 and 107 the authors comment that ‘Any discrepancies between reviewers were solved by consulting with a senior member.’ What do they mean by senior member?
In the discussion section, the authors rightly include the limitations of the study. However, the third limitation is not sufficiently clear. It would be interesting if the authors could clarify this issue (lines 294 to 297).
Author Response
1. Summary We thank the reviewers for their insightful feedback on our manuscript which will help us improve the quality of our work. 2. Point-by-point response to Comments and Suggestions for Authors
Comments 1: Methodologically the study is well supported and explained. The section of Materials and methods is very well explained. However, it would be interesting to include the keywords used in the search criteria to facilitate the reproducibility of the study.
Response 1: Thank you for the feedback. We agree that well-chosen keywords enable us to identify relevant articles easily. We have included a full list of search terms used for the PubMed database in the supplementary file, to facilitate reproducibility. The search strategy was developed using the key themes of the SMI population (including the subtypes, such as schizophrenia, bipolar disorders, other psychoses, and major depressive disorder), COVID-19 vaccination intervention, infection, hospital admissions, mortality outcomes, and observational studies.
Comments 2:
In line 106 and 107 the authors comment that ‘Any discrepancies between reviewers were solved by consulting with a senior member.’ What do they mean by senior member?
Response 2:
The phrase "senior member" refers to a senior researcher with extensive experience in conducting systematic reviews and meta-analyses in our research team. This individual served as a final arbiter for any discrepancies during the data extraction process, ensuring consistency and accuracy. To clarify this, we have revised the sentence to state the senior researcher's role and expertise explicitly.
On page 3, lines 112-114, the sentence “Any discrepancies between reviewers were solved by consulting with a senior member” is revised to “Any discrepancies between reviewers were solved by consulting with a senior researcher with extensive experience in conducting systematic reviews and meta-analyses within the research team”.
Comments 3:
In the discussion section, the authors rightly include the limitations of the study. However, the third limitation is not sufficiently clear. It would be interesting if the authors could clarify this issue (lines 294 to 297).
Response 3:
Thank you for the feedback. We agree to expand the third limitation to make it clearer. We have separated the third limitation into two sentences and provided an extended explanation:
“Third, only physical COVID-19-related outcomes in SMI patients were evaluated and did not take into consideration if there were any psychiatric-related adverse events following COVID-19 vaccination. Recent evidence from a large population-based cohort study in South Korea has shown that COVID-19 vaccination may increase the risks of certain psychiatric disorders, including depression, anxiety, and sleep disorders, while potentially reducing the risk of conditions like schizophrenia and bipolar disorder [64]. Evaluating the impact of COVID-19 vaccination on psychiatric outcomes in SMI patients can help to provide a comprehensive perspective on the effectiveness of COVID-19 vaccines. Fourth, due to the relatively short observation periods of the included studies, very few were able to report on the outcome of COVID-19-related mortality. In the future, prolonging the observation period will help capture the occurrence of mortality events.”
On page 10, lines 301-311, we have expanded the third limitation as described above.
Reviewer 4 Report
Comments and Suggestions for Authors
General comment
The paper “Effectiveness of COVID-19 Vaccines in People with Severe Mental Illness: A Systematic Review and Meta-analysis” is an interesting article about the evidence of the effectiveness of the COVID-19 vaccine in patients with severe mental illness (SMI) compared non-SMI controls. The article is well written and the results may be useful in improving knowledge and coverage of COVID-19 vaccination in patients with serious mental illness.
General comments
Some aspects should be explained in greater detail: for example, the use and utility of the rayyan program in this study and the flowchart that explains the sources and final selection of the 9 articles.
Specific comments
1) Abstract. In conclusion, the importance of psychiatrists and other medical specialists in improving vaccination coverage in these patients should be added..
2) Add flowchart that explains the sources and final selection of the 9 articles
3) Explain the usefulness of rayyan in this study.
4) Correct some errors (i.e: line 75 “..-, as December 2020 was the month when the first countries; Israel, the UK, and Denmark began population-level vaccination programs[31, 36]”; table 3 “ COVID-19-reltaed”).
5) Comment on the role of psychiatrists and other medical specialists in improving vaccination coverage and use of non-pharmaceutical measures in these patients
6) Check the reference style (ie: ref 28)
Author Response
1. Summary Thank you very much for taking the time to review this manuscript. We appreciate the reviewer's thoughtful comments and suggestions, which will help us further improve our manuscript. We have addressed each point as follows. 2. Point-by-point response to Comments and Suggestions for Authors
Comments 1: Abstract. In conclusion, the importance of psychiatrists and other medical specialists in improving vaccination coverage in these patients should be added.
Response 1: Thanks for your suggestion.
On page 1, lines 22-25, we have changed the last sentence in the Abstract section to “Vaccination appears an effective approach to prevent severe COVID-19 illness in SMI patients and actions should be taken by healthcare providers to improve vaccination coverage in these vulnerable populations.”
Comments 2: Add a flowchart that explains the sources and final selection of the 9 articles
Response 2: Thank you for your suggestion, the flowchart was included in the supplementary. To make this easier for the reader to access, we have moved this flow chart from the supplementary file to the main manuscript and named Figure 1 on page 4. Other picture numbers are changed in sequence.
Comments 3: Explain the usefulness of rayyan in this study.
Response 3: Thank you for your suggestion. Rayyan is a collaborative systematic review tool, designed for researchers conducting systematic reviews, meta-analyses, and literature reviews. It facilitates the screening, selection, and organization of relevant studies from various databases. We have added a sentence to the Methods section explaining the usefulness of Rayyan in our study.
On page 2, lines 87-89 “Using Rayyan facilitated the independent screening and selection of relevant studies for the reviewers, streamlining the process and enhancing efficiency.”
Comments 4: Correct some errors (i.e: line 75 “..-, as December 2020 was the month when the first countries; Israel, the UK, and Denmark began population-level vaccination programs[31, 36]”; table 3 “ COVID-19-reltaed”).
Response 4: Thank you for pointing this out. We have carefully reviewed the manuscript for grammatical and punctuation errors and made the necessary corrections.
On page 2, lines 79-81: the sentence “We restricted articles to those published after 1 December 2020, as December 2020 was the month when the first countries; Israel, the UK, and Denmark began population-level vaccination programs” is revised to “We restricted articles to those published after 1 December 2020, as December 2020 was the earliest time to initiate vaccination programs at the population-level”.
On page 7, table 2, the spelling error "COVID-19-reltaed" has been changed "COVID-19-related."
Comments 5: Comment on the role of psychiatrists and other medical specialists in improving vaccination coverage and use of non-pharmaceutical measures in these patients.
Response 5: Thanks for your suggestion.
On page 10, lines 314-323, we have revised the last paragraph of the Discussion section to highlight the role of healthcare providers in improving vaccination coverage and use of non-pharmaceutical measures in these patients. “Medical specialists, especially psychiatrists play a positive important role in addressing vaccine hesitancy or refusal of mental illness patients and ensuring equitable access to the COVID-19 vaccine [64]. They can provide SMI patients with information, address negative attitudes, and discuss the advantages along with possible risks of vaccination [29]. Our findings provide healthcare providers further evidence for improving vaccination coverage within this vulnerable group. Although vaccination has substantially reduced the risk of COVID-19-related outcomes, residual increased risks persist in patients with SMI. Proactively communicating with SMI patients and their caregivers about the increased risk for those patients can encourage further preventive steps for infection or monitoring for severe illness if infected.”
Comments 6: Check the reference style (ie: ref 28)
Response 6: Thank you for pointing out this. We have checked the reference style and revised the errors in ref 28, 56.
Round 2
Reviewer 1 Report
Comments and Suggestions for Authors
Dear authors,
Thank you for your answer, you did a lot of work.
All issues are solved, I have no remarks.